# Diversity of post-translational modifications and cell signaling revealed by single cell and single organelle mass spectrometry

Dong-Gi Mun[1], Firdous A. Bhat [1], Neha Joshi [1,2], Leticia Sandoval[1,3], Husheng Ding[1], Anu Jain[1], Jane A. Peterson[4], Taewook Kang [1], Ganesh P. Pujari[1], Jennifer L. Tomlinson[5], Rohit Budhraja [1], Roman M. Zenka [4], Nagarajan Kannan [1], Benjamin R. Kipp[1], Surendra Dasari[6], Alexandre Gaspar-Maia [1,3], Rory L. Smoot[5,7], Richard K. Kandasamy [1,6] ✉ & Akhilesh Pandey [1,2,3] ✉

The rapid evolution of mass spectrometry-based single-cell proteomics now enables the cataloging of several thousand proteins from single cells. We investigated whether we could discover cellular heterogeneity beyond proteome, encompassing post-translational modifications (PTM), protein-protein interaction, and variants. By optimizing the mass spectrometry data interpretation strategy to enable the detection of PTMs and variants, we have generated a high-definition dataset of single-cell and nuclear proteomic-states. The data demonstrate the heterogeneity of cell-states and signaling dependencies at the single-cell level and reveal epigenetic drug-induced changes in single nuclei. This approach enables the exploration of previously uncharted single-cell and organellar proteomes revealing molecular characteristics that are inaccessible through RNA profiling.

Unbiased measurement of the proteome from single cells has been facilitated by recent advancements in sophisticated sample preparation strategies using miniaturized and microfluidic devices[1–3]. As examples, nanoPOTS (Nanodroplet Processing in One pot for Trace Samples), cellenONE, and integrative proteomics chip have now enabled single cell isolation, lysis and digestion in nanoliter volumes of liquids, which minimize sample losses during the processes[4–6]. Simultaneously, there have been improvements in the performance of mass spectrometers[7] along with the development of novel methodologies to analyze single cells with enhanced sensitivity, throughput, and robustness[8–13]. In light of these developments, mass spectrometry-based single-cell proteomics now provides valuable insights into cellular heterogeneity by measuring proteome of individual cells[9,14,15]. In addition, the improved depth of proteome coverage allows the identification of post-translationally modified peptides from abundant proteins in single cells both in label-free and tandem mass tag (TMT)-based labeling strategies. For example, Orsburn and colleagues recently reported a TMT 9-plex labeling approach by spiking 50 ng of carrier protein to detect multiple PTMs in single cells[10]. Our own group also demonstrated the feasibility of

detecting phosphorylation and acetylation at a single-cell resolution in label-free mode by optimizing settings for trapped ion mobility spectrometry (TIMS)[16]. The benefits of a data-independent acquisition (DIA) approach for the detection of phosphorylation, acetylation, and ubiquitylation sites from low input samples have also been described although the samples were not strictly single cells but protein amounts diluted to near-single-cell equivalents[17].

Single-nucleus RNA sequencing has now become an alternative and complementary approach to single-cell RNA sequencing, showing advantages for transcriptomic profiling of samples that are difficult to generate high-quality single cell suspension[18,19]. However, the development of corresponding technologies for unbiased proteome profiling of single nuclei has been limited. Single nuclei yield a smaller amount of protein than single cells, which creates challenges for processing and analysis using mass spectrometry, where amplification is not feasible. Remarkably, before the recent introduction of highly sensitive mass spectrometers optimized for low-input samples, the feasibility of measuring metabolites from single cells and single organelles had been established using MALDI or capillary

[1]Department of Laboratory Medicine and Pathology, Mayo Clinic, Rochester, MN, 55905, USA. [2]Manipal Academy of Higher Education, Manipal, 576104 Karnataka, India. [3]Center for Individualized Medicine, Mayo Clinic, Rochester, MN, 55905, USA. [4]Proteomics Core, Mayo Clinic, Rochester, MN, 55905, USA. [5]Department of Surgery, Mayo Clinic, Rochester, MN, 55905, USA. [6]Department of Quantitative Health Sciences, Mayo Clinic, Rochester, MN, 55905, USA. [7]Department of Biochemistry and Molecular Biology, Mayo Clinic, Rochester, MN, USA. ✉e-mail: kandasamy.richardkumaran@mayo.edu; pandey.akhilesh@mayo.edu

electrophoresis coupled with ESI[20–22]. Most recently, a combination of deep ultraviolet laser ablation with nanodroplet sample handling showed the feasibility of proteome profiling from subcellular regions[23].

As the performance of single-cell proteomics continues to evolve, we explored the possibility of detecting additional types of PTMs. To this end, we performed single-cell proteome and PTM profiling using human normal cholangiocyte cell line and cholangiocarcinoma cell lines. As it was known that somatic mutations such as KRAS G12D and TP53 R175H are frequent in these cancer cell lines, we further tested the possibility of detecting variant peptides at single cell resolution. To this end, we established a strategy considering PTMs and variant peptides during interpretation of mass spectrometry data of single cells. This approach resulted in the identification of modified peptides including phosphorylation, methylation, and acetylation. Importantly, we identified peptides with variants from single cells. Further, we expanded our scope to include proteome and PTM profiling of single nuclei through which we measured expected drug-induced epigenetic changes. Overall, we demonstrated the significant role of single-cell and single-nuclei proteomics to understand cellular heterogeneity including PTMs and variants which cannot be achieved through single-cell genomics.

## Results

### Single-cell proteomics of cholangiocarcinoma cell lines

Based on a recent study demonstrating data independent acquisition (DIA) parallel accumulation-serial fragmentation (diaPASEF) mode to be superior for obtaining increased depth of proteome coverage at the single cell level[7], we adopted diaPASEF approach for single-cell proteomics of this study. Although multiple studies have highlighted the impact of spectral library composition on the overall performance of identification, especially when analyzing DIA data of low-input or single-cell samples using orbitrap mass spectrometry[17,24], there is no systematic study investigating the influence of spectral libraries on identification using timsTOF mass spectrometer. Thus, we first performed experiments using diluted peptides to determine the optimal strategy for analyzing diaPASEF data. Several spectral libraries varying in size were generated from DDA-PASEF data acquired from different amounts of HeLa protein digests and diaPASEF data of single-cell equivalent peptides were analyzed against these spectral libraries (Fig. 1a, S1a). In agreement with earlier studies[17,24], we observed that the large size of spectral library generated from >10 ng of peptides does not correspond to greater protein identification for low input samples (Fig. 1b). Interestingly, using spectral libraries generated from a lower amount of peptides (2–5 ng) resulted in achieving the maximum proteome coverage in

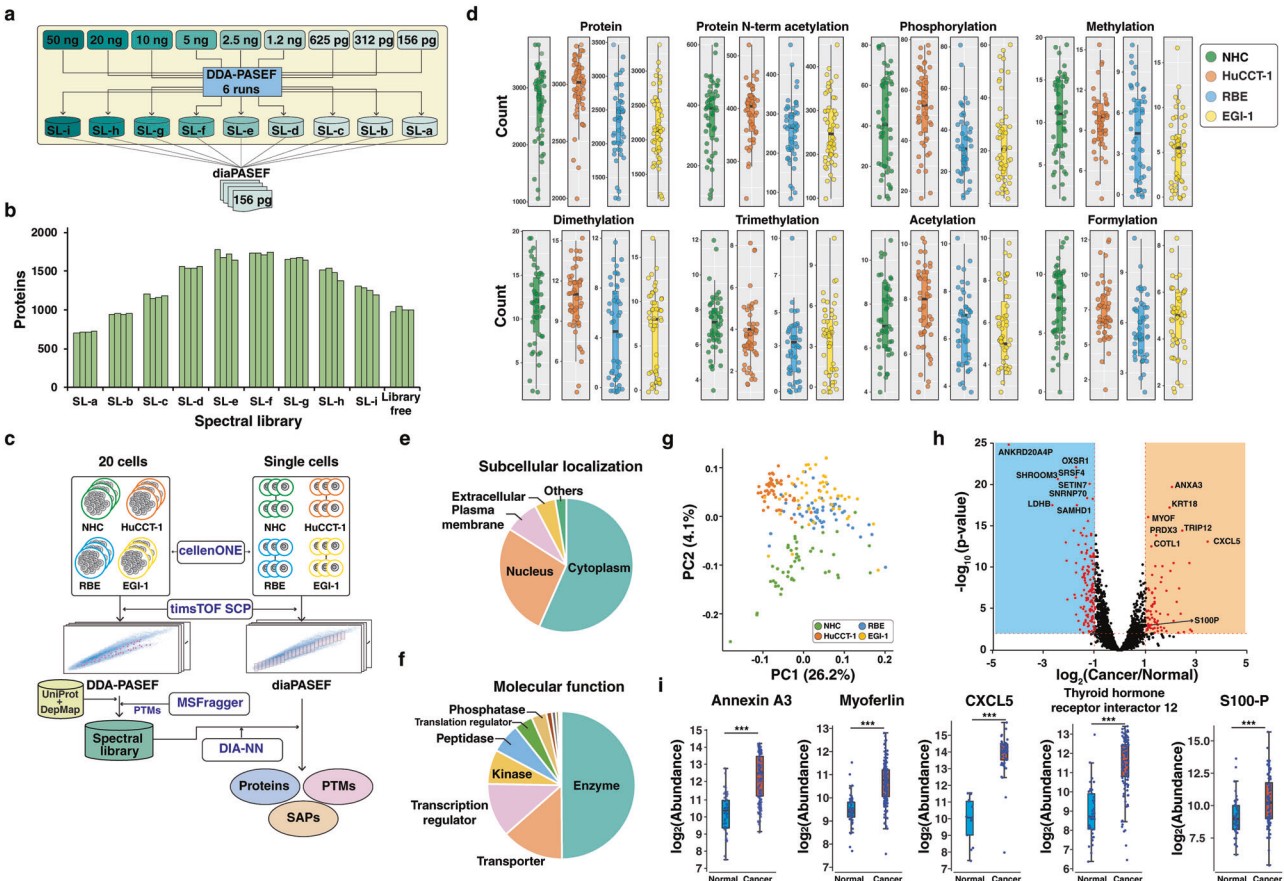

**Fig. 1 | Single cell proteomics of normal cholangiocytes and cholangiocarcinoma cells.** **a** Overall workflow for generating spectral libraries of different sizes through injection of different amounts of HeLa peptides as indicated and diaPASEF data from single cell-equivalent peptides. **b** Number of proteins identified from diaPASEF data of 156 pg peptides using DIA-NN. **c** Overall workflow for mass spectrometry-based proteome profiling of single cells. Cells were sorted using cellenONE platform and subsequent analyses were performed employing timsTOF SCP mass spectrometer. **d** Summary of the number of identified proteins, phosphorylated, protein N-terminal acetylated, lysine mono-, di-, trimethylated, acetylated, and formylated peptides from single cells. **e** Subcellular localization of identified proteins from single cells. **f** Molecular function of identified proteins.

Proteins categorized as other molecular types (1442 proteins) were excluded from the plot of molecular function. Ingenuity pathway analysis (QIAGEN Inc., https://digitalinsights.qiagen.com/IPA) was used for categorizing subcellular localization and function of proteins. **g** Principal component analysis of 4197 proteins. Each circle represents data from an individual cell. **h** Volcano plot showing fold-changes of 3,385 proteins quantified from ≥25% samples. Differentially expressed proteins are depicted in red. **i** Box plots of relative abundance of representative up-regulated proteins in cancer cells ($n = 150$) compared to normal cells ($n = 50$). The line inside the box represents the median, the lower and upper edges represent the first and third quartiles and the whiskers represent 1.5 times the interquartile range. *** Adjusted $p$-value ≤ 0.001.

both DIA-NN and Spectronaut (Figure. S1b). In addition, spectral library-based approach identified a significantly higher number of proteins compared to the spectral library-free approach. Thus, we decided to use peptides from ~20-50 cells or equivalent (2–5 ng peptides) for the generation of a spectral library, which was subsequently used to analyze the single-cell DIA mass spectrometry data.

To test if our improved mass spectrometry data analysis strategies could provide comprehensive insights on protein identifications and PTMs at the single cell level, we performed proteome profiling of single cells of normal human cholangiocytes cell line (NHC) and three cholangiocarcinoma cell lines (HuCCT-1, RBE and EGI-1) (Fig. 1c). Sorted single cells were analyzed using diaPASEF mode. Twenty cells were collected and triplicates of DDA-PASEF data were acquired to generate a spectral library. We considered protein N-terminal acetylation, phosphorylation on serine, threonine, and tyrosine, lysine methylation, dimethylation, trimethylation, acetylation, and formylation as variable modifications in addition to methionine oxidation, which were observed as major PTMs in open search strategy of FragPipe[25]. Additionally, we also tested the feasibility of detecting peptides with single amino acid polymorphisms (SAP) at the single cell level by incorporating the publicly available somatic single nucleotide variant (SNP) database (depmap.org) of these three cancer cell lines. This resulted in the identification of 50,121 peptides (4584 proteins) including 1012 modified peptides and 7 peptides with SAP, which were used for the generation of the spectral library through EasyPQP pipeline[26] (Fig. S2). Peptides with SAP detected from DDA-PASEF runs were confirmed by MS/MS spectra of synthetic peptides (Fig. S3). Next, we analyzed diaPASEF data of single cells using the generated spectral library (Fig. 1c). A total of 4197 proteins (41,560 peptides) were identified from 200 single cells with an average of 2548 proteins per single cell (Fig. 1d, Fig. S4a and Supplementary Data 1). The robustness of identification was assessed by checking the reproducibility of elution time and ion mobility of identified peptides, which resulted in a median coefficient of variation of 3.4% for elution time and 1.7% for ion mobility across 200 single-cell samples (Fig. S4b). Subcellular localization of proteins revealed that the majority of the identified proteins (56%) belong to cytoplasm followed by the nucleus (27%) and plasma membrane (8%) (Fig. 1e). In agreement with our previous observations[16], our current workflow for single cells is not optimized for detecting plasma membrane proteins, which requires further investigation. Of the proteins identified from single cells, enzymes constituted the majority (961 proteins), accompanied by 262 transporters, 227 transcriptional regulators, 145 kinases and 64 phosphatases (Fig. 1f). Principal component analysis revealed separation of normal cholangiocytes from cancer cells clusters of each cell line (Fig. 1g). Differential expression analysis showed that 84 proteins were upregulated, and 131 proteins were downregulated (|fold-change|>2 and adjusted $p$-value < 0.01) in the cancer cells as compared to the normal cells (Fig. 1h). Importantly, in this dataset, we identified molecules which are reported as potential markers for diagnosis hepato-pancreato-biliary cancer such as annexin A3[27] along with molecular progression aggressive clinical course including myoferlin[28,29]. Proteins related to prognosis and overall survival such as C-X-C motif chemokine ligand 5[30] and thyroid hormone receptor interactor 12[31] were shown to be upregulated in cancer cells. Interestingly, we observed upregulation of protein S100-P, which is a known diagnostic marker of cholangiocarcinoma[32] (Fig. 1i). In addition, the recent single-cell transcriptomic analysis of cholangiocarcinoma revealed that S100-P is a discriminatory biomarker for two subtypes of intrahepatic cholangiocarcinoma, perihilar large duct type, and perihilar small duct type[33]. Taken together, the successful detection of these previous reported biomarkers in single-cell proteomics indicates the feasibility of applying single-cell proteomics as a platform for potential cancer diagnostics and elucidation of cellular heterogeneity in which single-cell RNA sequencing is currently actively used to understand inherent cellular heterogeneity of cholangiocarcinoma[33–35].

## Protein-protein interaction networks at single-cell resolution

Given that a large number of proteins were identified in individual cells, we performed network analysis to identify potential heterogeneity in signaling at single-cell level. To this end, we constructed a protein-protein interaction network involving kinases using the human protein interactome data from BioGRID database[36]. We assembled a protein kinase network using all proteins detected from each cell line, resulting in a network with the largest connected component of ~1250 nodes and ~2600 edges (Fig. 2a, S5). We then overlaid the number of cells in which a protein was detected and found that there was a large variation in the expression of kinases. In addition, when clustering analysis on each network was performed to find densely connected regions, we identified a subnetwork that included several hub proteins such as epidermal growth factor receptor (EGFR), casein kinase 2 alpha 1 (CSNK2A1) and SRSF protein kinase 1 and 2 (SPRK1/2), which were present in all four cell lines but with varied expression at a single cell level. When focusing on the core subnetwork, we observed key kinases with varying levels of expression in individual cells (Fig. 2b). For instance, as expected, the expression of tumor suppressor protein, TP53 was lower in three cancer cell lines compared to normal cholangiocyte cells. Similarly, RAF proto-oncogene serine/threonine-protein kinase (RAF1) is significantly active in a majority of the HuCCT-1 cells but not in EGI-1, and cyclin-dependent kinase 6 (CDK6) was detected in only a fraction of cells in all four cell types. These data demonstrate that single-cell proteomics allows us to capture the heterogeneity of kinase abundance in single cells.

## Portion of housekeeping and non-housekeeping proteins detected in single cells

Next, we investigated if the protein expression signals measured at the single-cell level were sufficient to capture the overall cellular state including the basic cellular functions. Given that our network analysis revealed cellular heterogeneity at the level of signaling molecules, we checked the portion of housekeeping proteins detected per single cell and their corresponding relative abundance to verify that our analytical approach was able to overcome the challenge of the dynamic range of proteome and could detect proteins beyond the house-keeping functions that could truly indicate the cellulate state. We expected that a large majority of proteins detected in single cells were abundant housekeeping proteins. However, when compared to a publicly available database of housekeeping genes, HRT Atlas[37], only 13% of identified proteins were categorized as housekeeping proteins and were detected reproducibly across the 200 cells (Fig. S6a) while the large majority were not housekeeping proteins highlighting the utility of single-cell proteomics approach (Fig. S6b, c).

## Identification of SAPs at single cell resolution

Direct detection of mutant proteins resulting from somatic mutations provides compelling evidence of tumor and affords valuable insights into the mechanisms underlying tumorigenesis. Despite advancements in scRNA sequencing, the detection of somatic mutations at the single cell level remains as a major challenge[38,39]. Thus far, there is no report describing detection of proteins carrying variant sequences arising from mutations from single-cell proteomics. We hypothesized that improved proteome coverage of single cells could increase the likelihood of detecting mutations at the single cell level. To this end, we applied an approach that considered peptides with SAPs during the analysis of diaPASEF data (Fig. 1c). With this approach, we successfully identified peptides containing KRAS variants corresponding to G12D along with three other variants - IQGAP1 G1047R, CCT8 A488T, and GMPS R435T - at the single cell level (Fig. 3a and Supplementary Data 2). We confirmed the identification of these peptides by analyzing a mixture of synthetic peptides and assessing the similarity of elution time and fragmentation patterns (Figs. 3b, c). Importantly, peptide LVVVGADGVGK with KRAS G12D variant was only detected in HuCCT-1 and EGI-1 cells, not in RBE cells, which aligns with the genomic sequencing results annotated in DepMap. Similarly, peptide NVGLDIEAEVPTVK derived from CCT8 A4888T variant was only detected in RBE cells, which again agreed with data in DepMap. Finally, the simultaneous detection of peptides corresponding to non-mutated sequences indicates the feasibility of measuring relative levels of abnormal and normal proteins in each individual single cell. We anticipate that

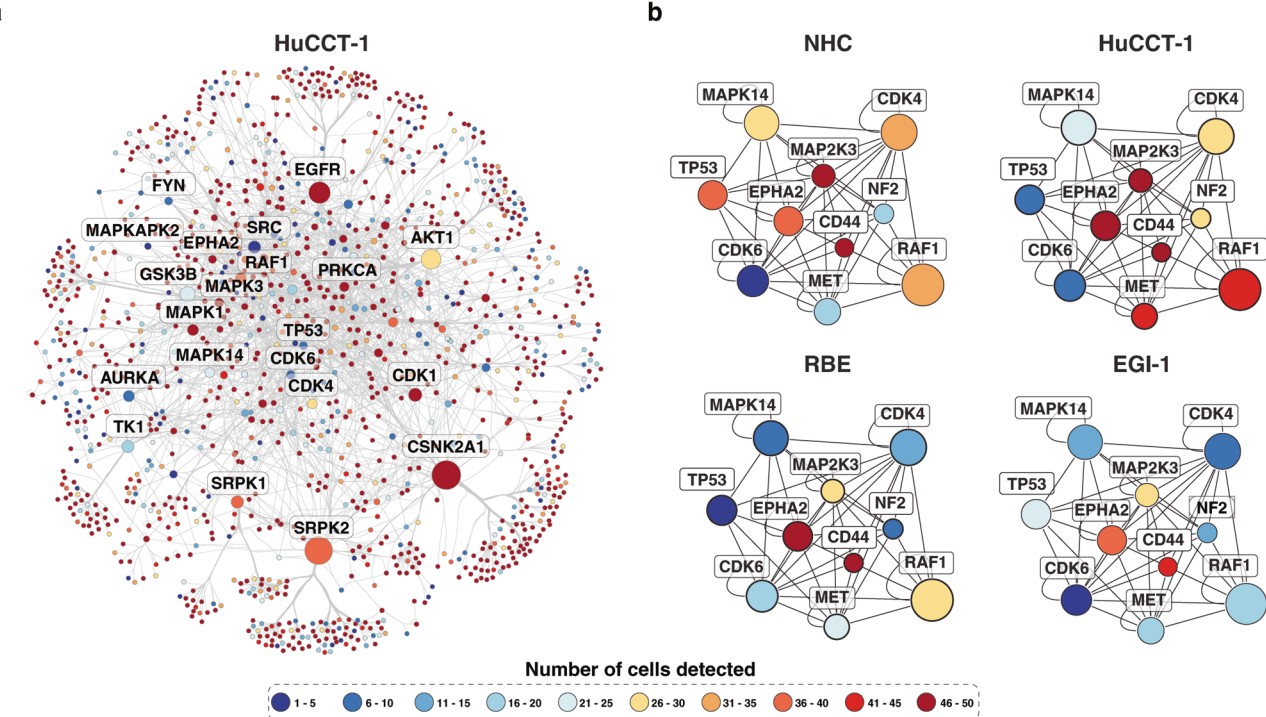

**Fig. 2 | Protein interaction networks at single-cell resolution. a** Protein-protein interaction map of proteins identified at single cells of HuCCT-1 cell line. **b** Core subnetwork showing key kinases with varying levels of expression in individual cells.

peptides arising from other types of variants such as indels and fusion genes could potentially be detected as the depth of proteome coverage of single cells continues to evolve.

## Identification of PTMs at single cell resolution

Our approach, which includes PTMs in the spectral library, enabled the identification of a total of 192 phosphorylated peptides (182 sites of 116 proteins), 24 lysine methylated peptides (24 sites of 19 proteins), 20 lysine dimethylated peptides (17 sites of 13 proteins), 14 lysine trimethylated peptides (11 sites of 10 proteins), 17 lysine acetylated peptides (14 sites of 13 proteins) and 16 lysine formylated peptides (16 sites of 12 proteins) (Fig. 1c and Supplementary Data 3). As expected, many of these PTMs were detected on abundant proteins based on the distribution of intensity-based absolute quantification (iBAQ) value of proteins with PTMs (Fig. S4c). Notably, we were able to detect multiple phosphorylation sites of proteins such as prelamin-A/C and nucleolin. For example, four phosphorylation sites (Ser 67, Thr 76, Thr 121, and Ser 563) were identified for nucleolin and these identifications were confirmed by elution time and fragment ions of synthetic peptides (Fig. 4a). Multiple types of PTMs from proteins such as histones and elongation factor 1-alpha 1 were also detected. For instance, five types of PTMs including methylation, dimethylation, trimethylation, acetylation, formylation on lysines (K14, K23, K27 and K79) of histone 3.1 were detected (Fig. 4b). Histone H3 modifications are known to affect gene expression – for instance, H3K14 acetylation is associated with activation of gene expression[40] which is known to coexist and coupled with H3K23 acetylation[41]; H3K79 methylation is associated with transcribed regions of active genes[42,43]; whereas H3K27 methylation is associated silencing gene expression via proximity or looping[44] and H3K79 formylation has been proposed to potentially silence gene expression since it interferes with H3K79 methylation[45]. Given that we have histone H3 modifications data at a single cell level, which could potentially indicate the overall transcriptional activity, we computed the ratio of PTMs activating gene expression versus repressing gene expression (Fig. 4c). We observed a higher ratio potentially indicating a higher transcriptional activity in most cells, which corroborates with the fact that the cells under study are proliferating tumor cell lines[46,47].

This analysis shows the potential to uncover individual cell states driven by gene regulation and transcriptional activity at single-cell resolution using histone modifications as a proxy.

We observed altered phosphorylation of several proteins in cancer cells (Fig. 4d). Although a direct link to cancer was not described for all identified sites, site-specific regulation in specific contexts had been reported for some of the identified sites. For example, phosphorylation of triosephosphate isomerase (TPI1) on Ser 21 was observed to be increased in HUCCT-1 cells. The activity of TPI1 is regulated via phosphorylation at Ser21 by the salt inducible kinases (SIK) in an LKB1-dependent manner, which is believed to influence tumorigenesis[48]. Alpha-enolase is known as a multifunctional oncoprotein[49,50] and phosphorylation on Ser115 is related to the activity of serine/threonine-protein kinase ULK1/2[51]. Calnexin phosphorylation at Ser 583 was increased, which is related to the recruitment of calnexin to ER-membrane-bound ribosomes for quality control[52]. This again indicates the potential of our approach to measure phosphorylation states at individual sites on proteins that correlate with their function at single-cell resolution.

Because a number of signaling studies have been done in immune cells, we treated Jurkat T cells with a phosphatase inhibitor calyculin A (for 15 and 30 minutes) followed by single-cell proteomics to directly measure alterations in response to perturbations (Fig. S7). This resulted in the identification of a total of 9236 peptides (1580 proteins) including 39 phosphorylated peptides (Fig. 4e). As expected, upregulation of phosphorylation was observed under calyculin A treatment demonstrating the feasibility of this approach (Fig. 4f). This included phosphorylation of nucleolin, actin and histone proteins, which are phosphorylated by different kinases such as casein kinases and cyclin-dependent kinases.

## Profiling of the whole proteome and PTMs at single nuclei

We also tested the feasibility of performing single-organelle proteomics to capture subcellular proteome dynamics. Since we were able to quantify important regulatory nuclear proteins such as histones and their PTMs in single cells, we ventured to perform proteome profiling in single nuclei as this could potentially provide a snapshot of the transcriptional state of cells. To this end, nuclei were isolated following an optimized protocol for single

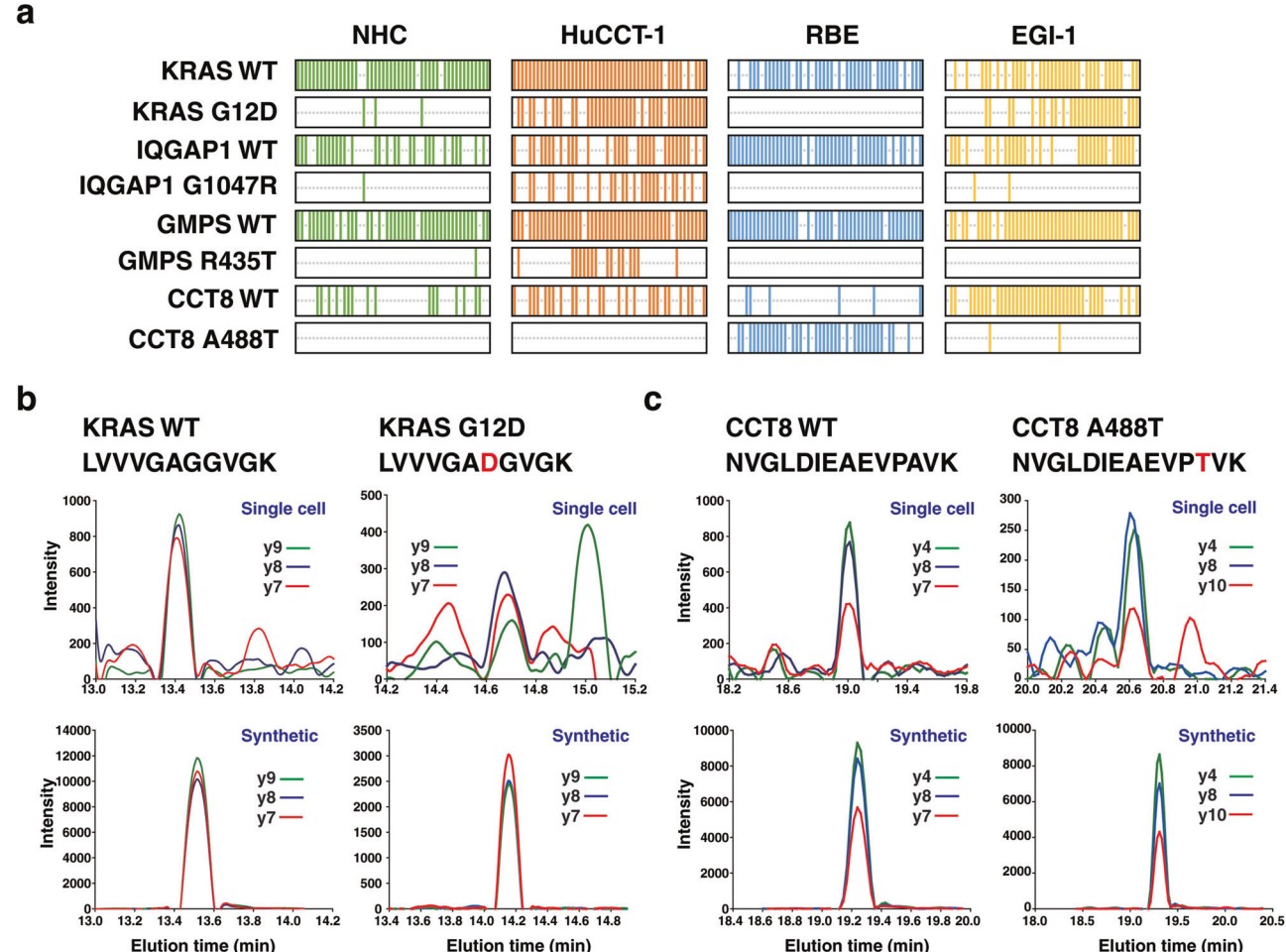

**Fig. 3 | Identification of SAPs at single cell resolution. a** Heatmap showing identification of peptides with SAPs and corresponding peptides with reference sequence across 200 single cell samples. **b** Extracted ion chromatograms of three fragment ions of KRAS wild type (WT) and G12D variant peptide. **c** Extracted ion chromatograms of three fragment ions of CCT8 WT and A488T variant peptide. Data of single cells and synthetic peptides are shown.

nuclei RNA sequencing[53] and sorted from a serous ovarian cancer cell line PEO1 treated with tazemetostat, a clinically approved epigenetic inhibitor of the histone methyltransferase EZH2 that methylates lysine 27 of histone 3.1[54] (Fig. 5a). We acquired diaPASEF data of single nuclei samples for both untreated and treated cells, which were interpreted using a spectral library generated from DDA-PASEF data of multiple nuclei. The same PTMs considered during experiments of single cells were included when generating the spectral library of nuclei. The spectral library generated from nuclei was composed of 14,238 peptides corresponding to 2094 proteins. When compared to the 4584 proteins in the spectral library of cells, 132 proteins were exclusively identified from the nuclei. As expected, proteins identified from nuclei were enriched for nuclear proteins, while proteins identified exclusively from cells were enriched in cytosol (Fig. 5b). Thirty single nuclei samples were analyzed including 15 from untreated and 15 from tazemetostat-treated cells, which resulted in a total of 6221 peptides corresponding to 1008 proteins with an average of 627 proteins per sample (Fig. 5c, S7c and Supplementary Data 4). Importantly, we were able to identify six different PTMs of histone H3.1 at K14, K23, K27 and K79, most of which were consistently detected in every single nuclei sample (Fig. 5d). The levels of trimethylation of H3K27 were reduced upon tazemetostat treatment, while there were no significant changes in the total protein abundance of histone H3.1 (Fig. 5e). Our study demonstrates the potential of single nuclear proteomics to study the molecular effects of drugs that can impact gene expression by measuring histone modifications.

## Discussion

In this study, we leveraged the advancements in single-cell proteomics to enable unbiased measurements of proteome and PTMs at the single-cell level. This revealed diversity in kinase expression within single cells offering insights into signaling heterogeneity. Further, our strategy allowed the detection of mutant proteins at the single-cell level, which we expect to be expanded for understanding tumor heterogeneity and evolution. Cholangiocarcinoma is an aggressive malignancy that has been studied at the single-cell level using a transcriptomic approach[34]. We believe that single-cell proteomics could also be used similarly to investigate the heterogeneity of cholangiocarcinoma to discover insights complementary to scRNA sequencing data. As techniques for single-cell proteomics continue to evolve, we believe that considering other PTMs and variants should be applied routinely to elucidate the cellular heterogeneity information, which is not obtainable through single-cell genomics. Further, it is imperative to continue improving detection sensitivity, particularly focusing on the consistent quantitation of PTMs across multiple single cells. As expected, the current detection of PTMs is biased towards abundant proteins. Additionally, the frequency of detection in 200 single-cell samples is also relatively low as compared to unmodified peptides. We anticipate that targeted approaches focused on specific PTMs should enhance reproducible measurements across single cells. The development of enrichment methods need to be investigated for PTMs such as phosphorylation by deploying microfluidic-based techniques.

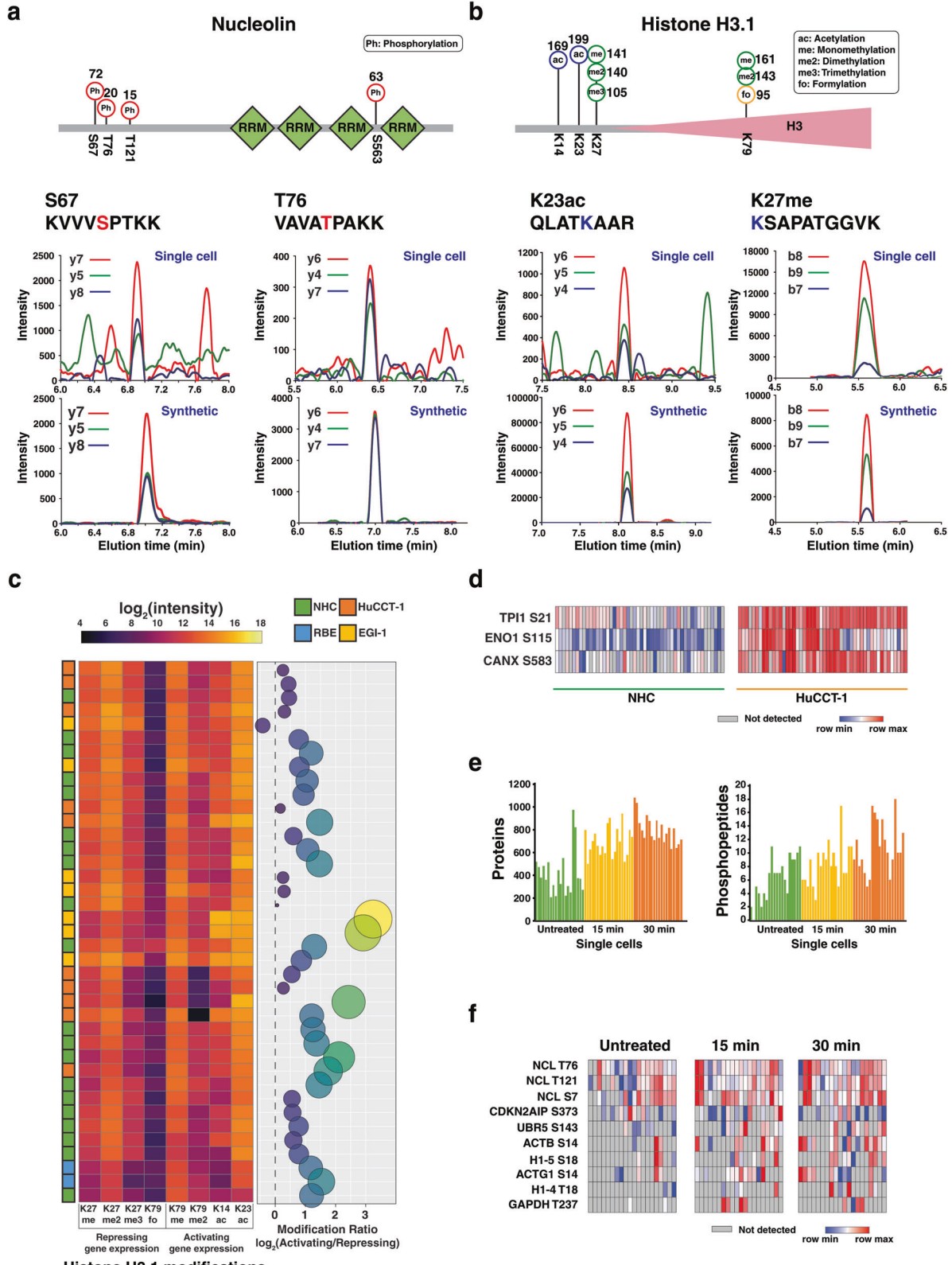

**Fig. 4 | Identification of PTMs in single cells. a** Identified phosphorylated sites of nucleolin (S67, T76, T121, and S563) from single cells. The number represents the count of detected samples out of 200 samples. Representative extracted ion chromatograms are shown both from single cells and synthetic peptides. **b** Histone H3.1 identified with PTMs of methylation, dimethylation, trimethylation, acetylation, and formylation on four lysines (K14, K23, K27, and K79). Representative extracted ion chromatograms are shown both from single cells and synthetic peptides. **c** Heatmap showing a subset of cells with levels of histone H3 PTMs that are activating (K79me/me2, K14ac and K23ac) or repressing (K27me/me2/me3 and K79fo) gene expression and a bubble plot depicting the ratio on a log_2 scale as indicated. A higher ratio indicates an overall increase in the level of transcriptional activity. **d** Heatmap showing phosphorylation sites that are upregulated (TPI S21, ENO1 S115, and CANX S583) in HuCCT-1 cells. **e** The number of identified proteins and phosphorylated peptides in Jurkat T cells of untreated and calyculin A treated conditions. **f** Heatmap showing the relative abundance of phosphorylation sites detected in untreated and calyculin A treated Jurkat cells.

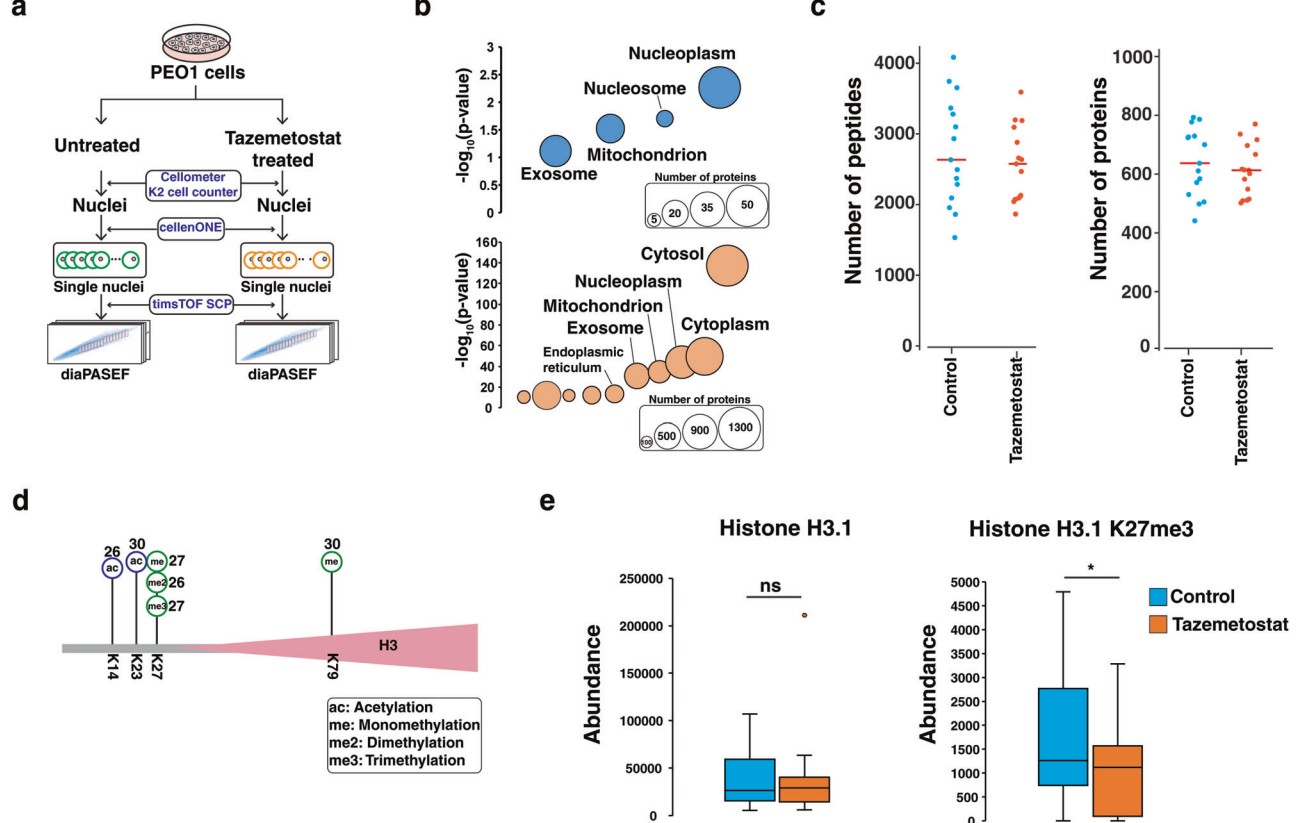

**Fig. 5 | Profiling of proteome and PTM of single nuclei. a** Schematic workflow for single nuclei proteome profiling from PEO1 cells. **b** Comparison of proteins from single cells and single nuclei. Cellular components for proteins exclusively identified in nuclei samples (132 proteins, blue circle) and proteins exclusively identified in cells (2622 proteins, orange circle) are shown. Gene ontology enrichment analysis was performed using DAVID. **c** The number of identified peptides and proteins across single nuclei samples from PEO1 cells in control and tazemetostat-treated conditions. **d** Histone H3.1 identified with PTMs of methylation, dimethylation, trimethylation, and acetylation on four lysines (K14, K23, K27 and K79). The number represents the count of detected samples out of 30 samples. **e** Relative abundance of histone 3.1 and trimethylation of K27 in control ($n = 15$) and tazemetostat-treated conditions ($n = 15$) shown as a boxplot with median, the lower and upper edges representing the first and third quartiles and the lower and upper whiskers represent the interquartile range× 1.5. ns: not significant; *$p < 0.05$.

In addition, by extending the scope to single nuclei, we demonstrated the power of technologies that can measure single organelle proteomes and offer critical insights into gene regulation such as drug-induced epigenetic changes. We isolated nuclei from cells using a widely adopted method for scRNA seq followed by single nuclei sorting on the CellenONE platform. We expect that alternative approaches using laser capture microdissection or ablation at a spatial resolution of <10 μm can also be deployed to isolate subcellular compartments[23,55]. Further, the emergence of innovative analytical methods other than mass spectrometry including electrode chemical analysis, super-resolution microscopy, and microfluidics has allowed qualitative and quantitative analysis to be carried out at single-cell and subcellular levels[56]. The integrative application of these technologies is expected to further expand our understanding of subcellular scale, even applicable to tissue samples[55,57].

## Methods
### Cell culture of normal cholangiocytes and cholangiocarcinoma cell lines
Normal human cholangiocytes (NHC) cell line and three cholangiocarcinoma cell lines (HuCCT-1, RBE and EGI-1) were kind gifts of Dr. Gregory Gores. The cell lines were cultured in the 10 cm dishes in RPMI-1640 medium (Gibco, 11875-093) containing 10% fetal bovine serum (FBS) (Gibco, 10437-028) with penicillin/streptomycin (Gibco, 15240-062). All cultures were maintained in a 5% $CO_2$ air-humidified atmosphere at 37 °C. For single-cell sorting, the media was removed, and the cells were gently washed with 10 ml 1xPBS (Corning, 21-040-CV)

twice. After wash, 0.05% trypsin-EDTA (Gibco, 25300-054) was added to the dishes and incubated for 5 minutes at 37°C. Trypsin digestion was stopped after adding 5 ml RPMI-1640 medium containing 10% FBS. The cells were spined down and washed with cold PBS twice. The cell density was counted by Invitrogen Countess II and adjusted to $2–5 × 10^5$ cells/ml.

### Cell culture of Jurkat cell line and calyculin A treatment
Jurkat cell line was maintained in RPMI-1640 medium (Gibco, 11875-093) containing 10% FBS (Gibco, 10437-028) with penicillin/streptomycin (Gibco, 15240-062). About $1 × 10^7$ Jurkat cells were untreated or treated with 0.1 μM calyculin A (Cell Signaling Technology, 9902) for 15 minutes and 30 minutes. Cells were harvested and washed with cold PBS twice for single-cell sorting. For the western blot, cells were lysed in modified RIPA buffer (50 mM Tris-HCl, pH 7.4, 150 mM NaCl, 1 mM EDTA, 1% Nonidet P-40, 0.25% sodium deoxycholate) followed by three cycles of sonication sonicated using a tip sonicator (Branson, SFX 550). Cell lysates were centrifuged at 12,000 g for 10 min at 4 °C and supernatants were transferred to a new tube. Protein estimation was performed using bicinchoninic acid protein assays (Thermo, 23225). Around 20 μg of proteins were resolved on SDS-PAGE gel, transferred to nitrocellulose membrane, and probed using a phospho-Ser/Thr antibody (Cell Signaling Technology, 9631) followed by reprobing with antibody against the corresponding protein. HSP90 antibody (Santa Cruz, sc-69703) was used as the loading control.

## Sample preparation of single cells

Single-cell sorting and reactions were performed using the cellenONE system (Cellenion, France)[6]. First, 1000 drops (~330 nl) of lysis buffer composed of 0.2% DDM (Millipore, 324355-1GM) and 100 mM TEAB (Sigma-Aldrich, T7408-500ML) were dispensed into each well of a 384-well plate. Cell suspension was loaded on the cellenONE system and cell dimensions including diameter and elongation factor were determined for single-cell sorting. The diameter for isolation was set as 25–30 µm and the maximum elongation factor was set as 1.97. Single cells were then isolated and deposited into the wells containing lysis buffer. Next, 1000 drops of buffer containing enzymes with the concentration of 2 ng/µl trypsin protease (Thermo Scientific, 90057) in 100 mM TEAB was dispensed into each well. The 384-well plate was incubated on the heating deck inside the cellenONE at 37 ℃ for 1 hour. The enzymaticreaction was quenched by adding 300 drops (~100 nl) of 5% formic acid. Centrifugation of the plate was done at 500 xg for 1 minute after each step of liquid dispensing to ensure liquid settles in the bottom of the wells. Digested samples from each well were reconstituted in 4 µl of 0.1% formic acid containing 0.05x iRT peptides (Biognosys, Ki-3002-1) and transferred to sample vials for mass spectrometry analysis. A lyophilized HeLa protein digest standard (Thermo Scientific, 88328) was used to prepare serially diluted peptides ranging from 50 ng to 156 pg.

## Sample preparation of single nuclei

PEO1 cell line was cultured in RPMI-1640 medium (Gibco, 11875-093) containing 10% FBS (Gibco, 10437-028) with penicillin/streptomycin (Gibco, 15240-062), 10 µg/ml insulin (Thermo, 12585014) and a 1:250 diluted nonessential amino acids (Gibco, 11130051) at 37 ℃ and 5% $CO_2$. PEO1 cells were treated with tazemetostat at a concentration of 30 µM with DMSO as vehicle or control for 72 hours. For nuclei isolation, 0.25% trypsin (Gibco, 25200072) was added and incubated at 37 ℃ and neutralized with media. The nuclei isolation was performed following the previously published protocol adapted from peripheral blood mononuclear cells (PBMC) nuclei isolation from the manufacture 10x genomics[53]. Briefly, about $1 \times 10^6$ cells were pelleted in a 2 ml microcentrifuge tube at 300 x $g$ for 5 min at 4 ℃ and resuspended in 100 µl chilled lysis buffer by pipetting 10 times. The cells were then incubated on ice for 3 min and, after the addition of 1 mL of wash buffer, they were centrifuged at 500 xg for 5 min at 4 ℃. The wash step was repeated one more time for a total of 2 washes. The pellet was resuspended in a chilled diluted Nuclei Buffer, and the nuclei concentration was assessed by propidium iodide (PI) staining (VitaStain) using a Cellometer K2 cell counter. Nuclei samples were loaded on cellenONE system to isolate into single nuclei under the diameter setting of 4.5–9 µm and the maximum elongation factor of 2.5. Lysis and tryptic digestion were performed following the same procedures of single cells as described in the previous section.

## Mass spectrometry data acquisition

Peptide samples of single cells were directly injected and separated on an analytical column (15 cm × 75 µm, $C_{18}$ 1.7 µm, IonOpticks, AUR3-15075C18-CSI) using nanoElute liquid chromatography system (Bruker Daltonics, Bremen, Germany). Solvent A (0.1% formic acid in water) and solvent B (0.1% formic acid in acetonitrile) were used to generate a linear gradient over 38 min; 5–30% solvent B in 20 min, 30–60% solvent B in 5 min, 60-80% solvent B in 3 min, maintaining at 80% solvent B for 5 min, and 5% solvent B for 5 min. The flow rate was set as 250 nl/min. Separated peptides were ionized using Captive spray source with a spray voltage of 1300 V and introduced into timsTOF SCP mass spectrometer (Bruker Daltonics, Bremen, Germany). For DDA-PASEF experiment, 10 PASEF scans were acquired with mass range of 100–1,700 m/z and ion mobility range of 0.7–1.3 Vscm$^{-2}$. Ion accumulation and ramp time were set as 180 ms, which was determined as an ideal setting for analyzing single cells[16]. The collision energy was linearly increased from 20 eV (0.6 Vscm$^{-2}$) to 59 eV (1.6 Vscm$^{-2}$). For diaPASEF experiment, ions were monitored in the range of 400–1000 m/z with an isolation window of 25 m/z and 8 PASEF scans per cycle along with 3 steps per PASEF scan. Ion mobility range was set at 0.64–1.37 Vscm$^{-2}$. To determine the optimal approach for analyzing diaPASEF mass spectrometry data, diluted peptide samples were prepared using HeLa protein digest standard (Thermo Scientific, 88328) and analyzed on an analytical column (8 cm × 50 µm, $C_{18}$ 1.5 µm, Bruker Daltonics) using the following gradients: 2–35% sol B in 20 min, 35–80% sol B in 3 min, maintaining at 80% sol B for 5 min and 2% sol B for 2 min.

## Data analysis

The raw mass spectrometry data were searched against UniProt human protein database (20,430 entries) using MSFragger (version 3.4) embedded in Fragpipe suite (version 17.0). When analyzing samples of diluted HeLa peptides, carbamidomethylation of cysteine was considered as a fixed modification, and oxidation of methionine and acetylation of protein N-terminal were set as variable modifications. For analyzing data of single cells, oxidation of methionine, acetylation of protein N-terminal/lysine, phosphorylation of serine/threonine/tyrosine, and methylation, dimethylation, trimethylation, formylation on lysine were considered as variable modifications. Somatic SNP database of HuCCT-1, RBE, and EGI-1 cell lines was downloaded from DepMap (https://depmap.org/portal/download/all/) and appended to the UniProt protein database considering two missed cleavage in both directions from the changed amino acid. DIA mass spectrometry data were analyzed in DIA-NN (version 1.8) using the following settings: network classifier = single-pass mode, protein inference = genes, quantification strategy = Robust LC (high accuracy), cross-run normalization = global and speed and RAM usage = optimal results. Spectronaut (version 16, Biognosys) was used to analyze diluted HeLa protein digests to determine the optimal spectral library for analyzing single-cell samples. For library free analysis, UniProt protein database was used for both of DIA-NN and Spectronaut. Some of the data analysis was performed in R (version 4.3.1) and plotting was done using the ggplot2 package. Network analysis and visualization of the protein-protein interactions were performed using Cytoscape (3.9.1). Protein-protein interaction data was downloaded from BioGRID database (release 4.4.222). Heatmaps were generated using Morpheus (https://software.broadinstitute.org/morpheus).

## Statistics and reproducibility

No data exclusion was performed in the data analysis. Non-supervised principal component analysis was performed to generate PCA plot. Student's t-test was used to calculate p-values, and the Benjamini-Hochberg correction was applied to calculate adjusted p-values.

## Peptide synthesis

The peptides were synthesized using standard FMOC chemistry on a MultiPep RSi (CEM Corp. Matthews) multiple peptide synthesizer at the 0.025 mmol scale. The starting resin for the light peptides was FMOC-Arg(pbf)-Wang resin or FMOC-Lys(Boc)-Wang resin (Novabiochem). For peptides with PTMs, the following derivatives were used: Fmoc-Ser(PO(OBzl)OH)-OH, Fmoc-Tyr(PO(OBzl)OH)-OH, and Fmoc-Thr(PO(OBzl)OH)-OH (Sigma-Aldrich) for phosphorylation, Fmoc-Lys(Ac)-OH (CreoSalus) for lysine acetylation, Fmoc-Lys(Me, Boc)-OH, Fmoc-Lys(Me)$_2$-OH, and Fmoc-Lys(Me$_3$Cl)-OH (Sigma-Aldrich) for lysine methylation, dimethylation, and trimethylation. The peptides were cleaved using the Razor cleaving apparatus (CEM Corp). Cleavage cocktail was trifluoroacetic acid, water, triisopropylsilane and 3,6-dioxa-1,8-octanedithiol (92.5/2.5/2.5/2.5 v/v/v/v). Peptides were precipitated and washed in cold methyl tert-butyl ether. Each peptide was HPLC purified and its molecular weight was verified with mass spectrometry. Synthetic peptides (1 fmol) were spiked into 1 ng peptides of bovine serum albumin and analyzed in DDA-PASEF and diaPASEF modes.

## Reporting summary

Further information on research design is available in the Nature Portfolio Reporting Summary linked to this article.

## Data availability

All mass spectrometry data have been deposited to the ProteomeXchange Consortium via the PRIDE partner repository [58] with the data set identifier PXD044986. The source data for figures in the paper can be found in Supplementary Data 5 and 6.

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

## Acknowledgements

This work was supported in part by grants from NCI to A.P. (U01CA271410 and P30CA15083). A. G-M. is funded by the Ovarian Cancer Academy (Department of Defense). A.G-M. and N.K. received support from the Mayo Clinic Ovarian Cancer SPORE grant P50 CA136393 and the Mayo Clinic Breast Cancer SPORE grant P50 CA116201 from the National Institutes of Health.

## Author contributions

D.-G.M. B.R.K., and A.P. conceptualized the study. D.-G.M., A.G.M., and A.P. designed the experiment. L.S., H.D., and J.L.T. prepared cells and nuclei samples. D.-G.M., F.A.B., and N.J. performed single-cell and nuclei sample preparation. D.-G.M., and F.A.B. acquired mass spectrometry data. D.-G.M., R.K.K., T.K., R.B., R.M.Z., and S.D. performed data interpretation. J.A.P., and A.J. prepared samples of synthetic peptides. G.P.P., N.K., and R.L.S. provided critical inputs. D.-G.M., R.K.K., and A.P. wrote the manuscript with input from all authors.

## Competing interests

The authors declare no competing interests.
