## [Peer Review File · Communications Biology]

Reviewers' comments:

Reviewer #1 (Remarks to the Author):

Thank you for the opportunity to read and review the submission "Diversity of PTMs and cell signaling revealed by single cell and single organelle mass spectrometry". This is a somewhat understated manuscript that highlights multiple new advances in the emerging field of single cell proteomics (SCP) by mass spectrometry. Most notably, in my opinion, is the description of the application of SCP to single mammalian nuclei, a feat that has not, to my knowledge, been previously reported. In addition to this striking accomplishment, the authors describe the identification of a large number of PTMs from diaPASEF analysis of 200 single cancer cells and the identification of peptide variants.

The manuscript is well-constructed, the figures are impressively intuitive and it was simply a joy to read. As is, the work will be an extremely valuable contribution to the emergence of SCP. I only have minor comments and suggestions.

- 1) Honestly, I think "single nuclei" should be the title. The identification of nearly 1,000 proteins (600-ish on average?) of single nuclei by LCMS is an accomplishment. I would suggest highlighting this.
- 2) Introduction line 65: I think the reference you cited here should be qualified in some way. At no point in the study from Hui Zhang's group does the group identify PTMs from dilutions of proteomic digests at concentrations approaching the realistic peptide levels from single cells. I believe they go as low as 600 picograms from a large volume container of lysate. Given sample loss, even on your impressive CellenOne system, this would be closer to 5 single cell equivalents. Perhaps "protein amount diluted to NEAR single cell equivalents" or similar?
- 3) Introduction lines 80 – 82. I have some trouble following the logic here. From the first sentence to the "thus" conclusion in the second. I feel like a concept is missing here, grammatically. Minor grammatical alterations may make this easier to follow.
- 4) Results beginning at line 110. While I think that your methods for building spectral libraries for SCP are probably the best that I've seen, and one that I will be borrowing immediately, I think that Vadim Demichev's group is using a similar strategy and this should be noted <https://doi.org/10.1101/2022.10.31.514544>
- 5) Line 128, I would shorten this length to depmap.org
- 6) Line 139 and Figure S4B – do you think that the subcellular compartment is a reflection of the proteome itself? Or is this a reflection of the relatively kind lysis method employed in your prep? A discussion of the interpretation of this distribution may be helpful to readers. SCP papers are being read outside of the LCMS proteomics field.
- 7) Line 215. Nothing needs changed here. I'm just impressed that you thought to consider both modified and unmodified variants and that you took the time to confirm this with synthetic peptides. Bravo.
- 8) Line 242, I think some clarification is necessary here. Higher transcriptional activity – in relation to what metric? I think I see what you're getting to here, but I think a reference is necessary here.
- 9) Profiling of single nuclei – unless I missed it, and I did look for it, I can't find any indication of how many nuclei were analyzed. Please add some metrics here.
- 10) Discussion – line 307 – please remove the word "easily" as no single cell genomics technique can quantify protein PTMs directly.
- 11) Methods line 361 –362. Some grammatical improvements are necessary here. Maybe something as simple as "the 384 well plate" "the enzyme reaction". Extremely minor to improve readability.
- 12) Line 410 – The 180ms ramp time is a curious setting. Was this identified as ideal in your recent publication? I don't recall seeing that, but I think this should be explained.
- 13) Line 452, the comma is unnecessary.
- 14) There are missing details on how the figures were generated. What software? I presume, R, and if so, then what packages were used.

Reviewer #2 (Remarks to the Author):

This is a very well written and nicely executed scientific piece of work by Mun et al., that add strongly to the growing field of single cell proteomics. While the upfront prep and separation of single cells was accomplished using the more standard approaches in the field right now, the authors have chosen to focus this work on the computation analyses of the proteomics data to identify post-translational modifications and variants and take one of the deepest dives into single cell data to date. I have a few questions to clarify a few things, and also some further suggestions for the authors to consider, but overall feel this is a substantial advance that will impact the proteomics field and recommend publication with some minor comments.

1. For the database searches, many variable modifications were searched simultaneously. I wonder if the authors also tried to pare down the modifications in batches to see if there were less mismatched spectra. For example, could they search all the lysine modifications alone and see if they get the same results as with searching with all the modifications at once? I am just curious about this, as it is well known that single cell peptide MS/MS spectra look a little different than bulk MS/MS data.

2. The histone data is very exciting. Histone PTMs is something my lab has worked on for a long time, and there are some challenges there for quantifying the histone modification patterns from tryptic peptides. In particular with just a trypsin digest, one can generate many peptides containing the same modified residue, making quantification really difficult to account for all the same residue containing different peptides. How did the authors deal with this potential issue in their quantification?

3. Similarly, there are some histone peptides (especially on the core or C-terminal end of histone H3) that will more easily generate peptides, such as the peptide that contains H3K79. This mark also seemed to be a prominent modification identified, but do the authors feel this is really the case versus identifying a peptide that creates a perfect tryptic peptide compared to other peptides?

4. The data shown for the modified phosphopeptides and histone peptides in the single cell data versus the synthetic peptides in Figure 4 seem odd. The retention times of the synthetic peptides are off from the endogenous. I am assuming the synthetic peptides are not heavy labeled and thus run separately. The best way to do these experiments is to spike in a heavy labeled peptide into the endogenous sample, and that way the light and heavy peptides have the exact retention times. I am not sure I am asking the authors to redo this experiment this way, but can they comment more on why they feel designed the experiments in this manner, and how they know for sure these are all the same peptides, given they are only following 3 transitions?

Again great work,

Reviewer #3 (Remarks to the Author):

I apologize for the length of time it took me to review this manuscript, but I went through it several times since I wanted to be sure to be comfortable with my recommendation since I rarely do this.

Based on repeated reading and analysis of this excellent body of work, I recommend acceptance of the current version. I really could not find anything substantive to recommend or critique.

We thank the reviewers for their comments. We have now addressed all of their concerns in the revised manuscript and feel that the quality of the manuscript has been improved. Following is a point-by-point response to the issues raised by the reviewers:

Reviewer: 1

Comments to the Author

Thank you for the opportunity to read and review the submission “Diversity of PTMs and cell signaling revealed by single cell and single organelle mass spectrometry”. This is a somewhat understated manuscript that highlights multiple new advances in the emerging field of single cell proteomics (SCP) by mass spectrometry. Most notably, in my opinion, is the description of the application of SCP to single mammalian nuclei, a feat that has not, to my knowledge, been previously reported. In addition to this striking accomplishment, the authors describe the identification of a large number of PTMs from diaPASEF analysis of 200 single cancer cells and the identification of peptide variants.

The manuscript is well-constructed, the figures are impressively intuitive and it was simply a joy to read. As is, the work will be an extremely valuable contribution to the emergence of SCP. I only have minor comments and suggestions.

We thank the reviewer for appreciating the significance of our manuscript.

1. Honestly, I think “single nuclei” should be the title. The identification of nearly 1,000 proteins (600-ish on average?) of single nuclei by LCMS is an accomplishment. I would suggest highlighting this.

We thank the reviewer for the suggestion. We would like to use the term “single organelle” to highlight that this is the first demonstration for analysis of any single organelle.

2. Introduction line 65: I think the reference you cited here should be qualified in some way. At no point in the study from Hui Zhang’s group does the group identify PTMs from dilutions of proteomic digests at concentrations approaching the realistic peptide levels from single cells. I believe they go as low as 600 picograms from a large volume container of lysate. Given sample loss, even on your impressive CellenOne system, this would be closer to 5 single cell equivalents. Perhaps “protein amount diluted to NEAR single cell equivalents” or similar?

As the reviewer suggested, we have now revised the manuscript and it read “...diluted to near single cell equivalents”

3. Introduction lines 80 – 82. I have some trouble following the logic here. From the first sentence to the “thus” conclusion in the second. I feel like a concept is missing here, grammatically. Minor grammatical alterations may make this easier to follow.

We apologize if this was not clear. We have now revised the manuscript and it reads “As the performance of single cell proteomics continues to evolve, we explored the possibility of detecting additional types of PTMs. To this end, we performed single cell proteome and PTM profiling using human normal cholangiocyte cell line and cholangiocarcinoma cell lines.”

4. Results beginning at line 110. While I think that your methods for building spectral libraries for SCP are probably the best that I’ve seen, and one that I will be borrowing immediately, I think that Vadim Demichev’s group is using a similar strategy and this should be noted <https://doi.org/10.1101/2022.10.31.514544>

We thank the reviewer for pointing this out. We were aware of this study by Vadim Demichev's group, but were not able to cite since this is still in bioRxiv.

5. Line 128, I would shorten this length to depmap.org

We have updated this in the revised the manuscript.

6. Line 139 and Figure S4B – do you think that the subcellular compartment is a reflection of the proteome itself? Or is this a reflection of the relatively kind lysis method employed in your prep? A discussion of the interpretation of this distribution may be helpful to readers. SCP papers are being read outside of the LCMS proteomics field.

We thank the reviewer for bringing this to our notice. We concluded that our results do not reflect the entire proteome. When compared to the subcellular compartment of the whole proteins in the UniProt human protein database, we missed a portion of plasma membrane proteins. Please see the pie chart below, which shows the distribution of subcellular compartment of proteins obtained from single cell experiments and in the UniProt human protein database.

To increase the coverage of plasma membrane proteome, further investigation of single cell sample preparation will be required, which we also observed in our recent study (Mun, D.G., Bhat, F.A., Ding, H., Madden, B.J., Natesampillai, S., Badley, A.D., Johnson, K.L., Kelly, R.T., and Pandey, A., *Optimizing single cell proteomics using trapped ion mobility spectrometry for label-free experiments. Analyst, 2023. 148, 3466-75*).

We have now revised the manuscript and it reads “In agreement with our previous observations, our current workflow for single cells is not optimized for detecting plasma membrane proteins, which requires further investigation”.

7. Line 215. Nothing needs changed here. I’m just impressed that you thought to consider both modified and unmodified variants and that you took the time to confirm this with synthetic peptides. Bravo.

We thank the reviewer for this comment.

8. Line 242, I think some clarification is necessary here. Higher transcriptional activity – in relation to what metric? I think I see what you’re getting to here, but I think a reference is necessary here.

We thank the reviewer for bringing this to our attention. As mentioned in the manuscript, this is an inference we draw at the single cell level based on the histone H3 modifications that activate/repress gene expression. At a single cell level, the total abundance of histone H3 modifications activating gene expression was higher

than the total abundance of histone H3 modifications repressing gene expression. Thus, our deduction of “Higher transcriptional activity” is “high levels of active histone H3 modifications ~ higher gene expression.” We have now added proper references to make this conclusion.

9. Profiling of single nuclei – unless I missed it, and I did look for it, I can’t find any indication of how many nuclei were analyzed. Please add some metrics here.

We apologize if this was not clear. A total of 30 single nuclei samples were analyzed. We have now revised the manuscript and it reads “Thirty single nuclei samples were analyzed including 15 from untreated and 15 from tazemetostat-treated cells”.

10. Discussion – line 307 – please remove the word “easily” as no single cell genomics technique can quantify protein PTMs directly.

We agree with the reviewer that single cell genomics technique is not able to quantify PTMs directly. We have now removed the word “easily”.

11. Methods line 361 –362. Some grammatical improvements are necessary here. Maybe something as simple as “the 384 well plate” “the enzyme reaction”. Extremely minor to improve readability.

We have now updated the text as recommended.

12. Line 410 – The 180ms ramp time is a curious setting. Was this identified as ideal in your recent publication? I don’t recall seeing that, but I think this should be explained.

We thank the reviewer for bringing this to our notice. Yes, this is the ideal setting we determined for analyzing single cell samples in DDA-PASEF mode, and this finding was published in our previous publication (Mun, D.G., Bhat, F.A., Ding, H., Madden, B.J., Natesampillai, S., Badley, A.D., Johnson, K.L., Kelly, R.T., and Pandey, A., Optimizing single cell proteomics using trapped ion mobility spectrometry for label-free experiments. Analyst, 2023. 148, 3466-75). We have now revised the manuscript and it reads “Ion accumulation and ramp time were set as 180 ms, which was determined as an ideal setting for analyzing single cells.”

13. Line 452, the comma is unnecessary.

This has been updated now.

14. There are missing details on how the figures were generated. What software? I presume, R, and if so, then what packages were used.

We have now included the software used to generate figures in Method section.

Reviewer: 2

Comments to the Author

This is a very well written and nicely executed scientific piece of work by Mun et al., that add strongly to the growing field of single cell proteomics. While the upfront prep and separation of single cells was accomplished using the more standard approaches in the field right now, the authors have chosen to focus this work on the computation analyses of the proteomics data to identify post-translational modifications and variants and take one of the deepest dives into single cell data to date. I have a few questions to clarify

a few things, and also some further suggestions for the authors to consider, but overall feel this is a substantial advance that will impact the proteomics field and recommend publication with some minor comments.

1. For the database searches, many variable modifications were searched simultaneously. I wonder if the authors also tried to pare down the modifications in batches to see if there were less mismatched spectra. For example, could they search all the lysine modifications alone and see if they get the same results as with searching with all the modifications at once? I am just curious about this, as it is well known that single cell peptide MS/MS spectra look a little different than bulk MS/MS data.

The reviewer raises an excellent point. As the reviewer suggested, we performed a protein database search on one DDA-PASEF data considering one lysine PTM at a time using MSFragger search engine. Interestingly, several peptides with PTMs were additionally identified through this approach, as summarized in a table below.

	Total peptides	Methylated peptides	Dimethylated peptides	Trimethylated peptides	Acetylated peptides	Formylated peptides
All together	22,549	22	-	-	13	24
Methylation	22,486	21	-	-	-	-
Dimethylation	22,520	-	17	-	-	-
Trimethylation	22,387	-	-	9	-	-
Acetylation	22,514	-	-	-	8	-
Formylation	22,393	-	-	-	-	9

To the best of our knowledge, no one has addressed this in the single cell proteomics study. In addition, evaluating the effect of the way of considering PTMs on single cell DIA mass spectrometry data is surely interesting but is out of scope of our current study and will require a more detailed investigation.

2. The histone data is very exciting. Histone PTMs is something my lab has worked on for a long time, and there are some challenges there for quantifying the histone modification patterns from tryptic peptides. In particular with just a trypsin digest, one can generate many peptides containing the same modified residue, making quantification really difficult to account for all the same residue containing different peptides. How did the authors deal with this potential issue in their quantification?

We thank the reviewer for bringing this to our notice. We acknowledge that quantitation of PTMs especially for histones is difficult. For our study, we summed quantity of peptides containing the same modified residue. From our single cell experiment, 17 PTMs identified with multiple peptides including acetylation of H3K23 and H3K14, which is negligible portion compared to what others reported in the experiments from bulk samples.

3. Similarly, there are some histone peptides (especially on the core or C-terminal end of histone H3) that will more easily generate peptides, such as the peptide that contains H3K79. This mark also seemed to be a prominent modification identified, but do the authors feel this is really the case versus identifying a peptide that creates a perfect tryptic peptide compared to other peptides?

We agree with the reviewer and thank for suggesting great idea for the future study. We did not consider peptides with more than 2 missed cleavages during protein database search. We also did not consider using other enzymes such as Lys-C and Glu-C. To identify as many as PTMs of histone proteins in single cells as possible, further systematic investigation is required using the strategies which have been established by several groups, including Ben Garcia's group.

4. The data shown for the modified phosphopeptides and histone peptides in the single cell data versus the synthetic peptides in Figure 4 seem odd. The retention times of the synthetic peptides are off from the endogenous. I am assuming the synthetic peptides are not heavy labeled and thus run separately. The best way to do these experiments is to spike in a heavy labeled peptide into the endogenous sample, and that way the light and heavy peptides have the exact retention times. I am not sure I am asking the authors to redo this experiment this way, but can they comment more on why they feel designed the experiments in this manner, and how they know for sure these are all the same peptides, given they are only following 3 transitions?

The reviewer is correct. We agree that spiking heavy peptides into the single cell samples is the best way to confirm our findings at single cell resolution. However, we observed light contamination from stable isotope labeled peptides (>99% purity) incorporated with Lys8 or Arg10, which was reported in elsewhere (Salek, M., Forster, J.D., Lehmann, W.D., and Riemer, A.B., Light contamination in stable isotope-labelled internal peptide standards is frequent and a potential source of false discovery and quantitation error in proteomics. Analytical and Bioanalytical Chemistry, 2022. 414, 2545-52). In addition, we used a highly sensitive mass spectrometer, timsTOF SCP, which detects non-negligible signal of light contamination even when injecting <1 fmol of SIL peptides. This requires further careful investigation to find optimal spiking amount or to synthesize double isotopically labelled heavy peptides similar to what we used in our previous study (Renuse, S., Vanderboom, P.M., Maus, A.D., Kemp, J.V., Gurtner, K.M., Madugundu, A.K., Chavan, S., Peterson, J.A., Madden, B.J., Mangalparthi, K.K., Mun, D.G., Singh, S., Kipp, B.R., Dasari, S., Singh, R.J., Grebe, S.K., and Pandey, A., A mass spectrometry-based targeted assay for detection of SARS-CoV-2 antigen from clinical specimens. EBioMedicine, 2021. 69, 103465). This is why we analyzed light synthetic peptides separately for the study. As the reviewer is concerned, although we observed a slight shift in the retention time for several peptides, particularly QLATK(ac)AAR, we confirmed identification based on annotated MS/MS spectra of endogenous and synthetic peptides both in DDA-PASEF and diaPASEF experiments.

For clarification, we have now added the sentence and it reads as “Synthetic peptides (1 fmol) were spiked into 1 ng peptides of bovine serum albumin and analyzed in DDA-PASEF and diaPASEF modes”.

Reviewer: 3

Comments to the Author

I apologize for the length of time it took me to review this manuscript, but I went through it several times since I wanted to be sure to be comfortable with my recommendation since I rarely do this.

Based on repeated reading and analysis of this excellent body of work, I recommend acceptance of the current version. I really could not find anything substantive to recommend or critique.

We thank the reviewer for appreciating the significance of our manuscript.

REVIEWERS' COMMENTS:

Reviewer #1 (Remarks to the Author):

Thank you for your thoughtful responses to my and other reviewer comments and suggestions during the previous round of review. I can't see where anything was missed here.

Reviewer #2 (Remarks to the Author):

The comments from the authors are thoughtful and well described. I think they have in good faith answered the comments as best as they could, and added text to the manuscript when appropriate. I now highly support publication of this manuscript.